# A Systematic Review of Molecular Imaging Agents Targeting Bradykinin B1 and B2 Receptors

**DOI:** 10.3390/ph13080199

**Published:** 2020-08-17

**Authors:** Joseph Lau, Julie Rousseau, Daniel Kwon, François Bénard, Kuo-Shyan Lin

**Affiliations:** 1Department of Molecular Oncology, BC Cancer, Vancouver, BC V5Z 1L3 Canada; lauj2@nih.gov (J.L.); jrousseau@bccrc.ca (J.R.); dkwon@bccrc.ca (D.K.); 2Department of Radiology, University of British Columbia, Vancouver, BC V6T 1Z4, Canada

**Keywords:** kinins, bradykinin receptors, optical imaging, nuclear imaging, personalized medicine

## Abstract

Kinins, bradykinin and kallidin are vasoactive peptides that signal through the bradykinin B1 and B2 receptors (B1R and B2R). B2R is constitutively expressed in healthy tissues and mediates responses such as vasodilation, fluid balance and retention, smooth muscle contraction, and algesia, while B1R is absent in normal tissues and is induced by tissue trauma or inflammation. B2R is activated by kinins, while B1R is activated by kinins that lack the C-terminal arginine residue. Perturbations of the kinin system have been implicated in inflammation, chronic pain, vasculopathy, neuropathy, obesity, diabetes, and cancer. In general, excess activation and signaling of the kinin system lead to a pro-inflammatory state. Depending on the disease context, agonism or antagonism of the bradykinin receptors have been considered as therapeutic options. In this review, we summarize molecular imaging agents targeting these G protein-coupled receptors, including optical and radioactive probes that have been used to interrogate B1R/B2R expression at the cellular and anatomical levels, respectively. Several of these preclinical agents, described herein, have the potential to guide therapeutic interventions for these receptors.

## 1. Introduction

Bradykinin B1 and B2 receptors (B1R and B2R) are transmembrane receptors that belong to the rhodopsin-like G protein-coupled receptor (GPCR) superfamily [1,2]. In the 1970s, work by Regoli and colleagues led to the identification and characterization of both receptors, including their pharmacological profiles and expression patterns [3,4,5]. B1R is generally absent in healthy tissues, and its expression is induced by injury and inflammation [6,7]. On the other hand, B2R is ubiquitously expressed throughout the body and is involved in vasodilation, osmoregulation, smooth muscle contraction, and nociceptor activation [1]. B1R and B2R have emerged as therapeutic targets as they are implicated in inflammatory disease, vasculopathy, neuropathy, obesity, diabetes, and cancer [8]. B1R and B2R can hold dichotomous roles in diseases; thus, agonists and antagonists have been evaluated as therapeutics [1].

In the present review, we summarize molecular imaging (MI) agents targeting the B1R and B2R. MI is a research discipline that enables the visualization, characterization, and quantitation of biological processes at the cellular and/or anatomical levels [9,10,11]. MI is used to study conditions that underlie normal physiology and pathologies. MI modalities include, but are not limited to, fluorescence imaging, bioluminescence imaging, near-infrared imaging (NIR), single photon emission computed tomography (SPECT), positron emission tomography (PET), and magnetic resonance imaging [12,13,14,15]. While the sensitivity of these modalities varies, they can provide information of high spatiotemporal resolution. MI is already well-integrated in clinical practice for detecting disease occurrence, guiding therapeutic selection, and predicting and monitoring treatment response. Despite their promise, the development of MI probes targeting B1R and B2R is limited to preclinical settings.

## 2. Kinin Receptors, Signaling, and Ligands

The signaling of bradykinin receptors is mediated by kinins [16], which are produced when kininogens are cleaved by serine proteases called kallikreins (Figure 1a). This is also known as the kallikrein–kinin system (KKS) [17,18]. In mammals, kinins are peptides that include bradykinin (BK: Arg-Pro-Pro-Gly-Phe-Ser-Pro-Phe-Arg), kallidin (KD: Lys-BK), methionyl-lysyl-BK, and their C-terminal des-Arg derivatives [18]. Kininases I (carboxypeptidase-M and carboxypeptidase-N) are responsible for removing the carboxyl-Arg group [18]. Kinins can bind either receptor, but they demonstrate receptor selectivity. BK and KD bind preferentially to B2R, while des-Arg^9^-BK and Lys-des-Arg^9^-BK bind preferentially to B1R [1]. Kinins are also enzymatically regulated by kininase II (angiotensin I-converting enzyme), neprilysin, and endothelin-converting enzyme [18]. Cleavage by kininase II inactivates kinins, which have a biological half-life reportedly to be 30 s or less [8,19].

Following ligand-binding, B1R and B2R signal through associated G proteins to activate signaling molecules like protein kinase C and phospholipases, and secondary messengers like inositol-1,4,5,-triphosphate, diacylglycerol, calcium, and arachidonic acid (Figure 1b) [18]. These secondary messengers go on to modulate other signaling processes (e.g., nitric oxide or prostaglandin production) [18]. The biological consequence of B1R or B2R activation depends on the cell type. B1R and B2R can compensate for one another for signaling, as demonstrated in different knockout models [20,21]. An example is the coregulation of blood pressure in rodents [22]. However, due to the difference in expression levels between B1R (induced) and B2R (constitutive), B2R is the receptor that predominantly mediates the kinin response in the body. This poses unique implications for the development of therapeutic agents and MI probes, which we will discuss later.

The discovery of kinins by Rocha et al. preceded the isolation and characterization of B1R and B2R by several decades [16]. The BK peptide was the first peptide to be synthesized using the modern day solid-phase peptide synthesis strategy developed by Merrifield [23]. Since then, many kinin derivatives have been synthesized and investigated. These sequences include modifications (e.g., amino acid substitutions, reduction of amide bonds, *N*-terminal capping [24]) that are aimed at conferring selectivity, stability to peptidases, agonist/antagonist properties and prolonging in vivo pharmacological effects [25]. Moreover, the plasticity of kinins also enables their use as targeting vectors to carry fluorophores, drugs, and radioactivity (Figure 2). Icatibant, a B2R antagonist, is the only kinin to receive U.S. Food and Drugs Administration approval (Table 1 and Figure 3). It is indicated for the treatment of acute attacks of hereditary angioedema in adults with C1-esterase inhibitor deficiency [26].

Beyond peptides, there is continual interest in small molecule agonists and antagonists that target the bradykinin receptors [1,27,28,29,30] These small molecules were developed in part to improve oral bioavailability and brain penetrance. Early candidates were met with challenges like receptor (B1R vs. B2R) or species (rodent vs. human) specificity [31], but these were gradually addressed with new designs. Anatibant and fasitibant, B2R antagonists, advanced into clinical testing for treatment of traumatic brain injury and knee osteoarthritis, respectively [32,33]. However, their development was discontinued due to the lack of efficacy. The hope is that new scaffolds will be able to find clinical utility in other disease settings [27].

**Table 1 pharmaceuticals-13-00199-t001:** B1R and B2R targeting therapeutics in clinical trials. Adapted from da Costa, P.L.N., et al. [17].

Drug	Target	Clinical Phase	Indications	Comments	Reference
HOE-140 (Icatibant)	B2R antagonist	Approved	Hereditary angioedema	Shortened the duration of acute attacks.	[26]
		Phases I-IV	Cardiopulmonary bypass, inflammation, fibrinolysis, surgery, ischaemic heart diseases, ischaemic reperfusion, heart failure, ACE inhibitor associated angioedema, angioneurotic edema	Many completed and ongoing studies. Decreased intraoperative fibrinolytic capacity in cardiopulmonary bypass. No efficacy demonstrated for angioedema and ischemia-reperfusion injury.	[34,35,36]
		Phase II	Mitochondria and chronic kidney disease	Completed, no evidence of efficacy.	NCT03177798
		Phase II	Knee pain in osteoarthritis	Completed, results not available.	NCT00303056
MEN16132 (Fasitibant)	B2R antagonist	Phase II	Knee pain in osteoarthritis	Two studies completed. No direct evidence of efficacy, treated patients used less rescue medication.	NCT01091116 NCT02205814
CP-0127 (Deltibant)	B2R antagonist	Phase II	Severe traumatic brain injury sepsis	Ineffective for sepsis. Discontinued due to unexpected preclinical findings.	[37]
LF16-0687 (Anatibant)	B2R antagonist	Phase II	Severe traumatic brain injury	Inconclusive results and possible safety issues. Trial halted.	[32]
RMP-7 (Lobradimil)	B2R agonist	Phase II	Childhood brain tumors	Completed. No improved efficacy.	[38]
		Phase I	HIV infection and cryptococcal meningitis	Completed, results not available.	NCT00002316
FOV-2304 (Safotibant)	B1R antagonist	Phase II	Diabetic macular edema	Discontinued, results not available.	[17,39]
MK-0686	B1R antagonist	Phase II	Postherpetic neuralgia, postoperative dental pain, osteoarthritis	Terminated for postherpetic neuralgia, completed for dental pain and osteoarthritis. No results disclosed.	[17]
BI-113823	B1R antagonist	Phase I	Osteoarthritis	Terminated.	NCT01207973
SSR-240612	B1R antagonist	Phase II	Inflammation and neuropathic pain	Halted for undisclosed reasons.	[40]
B9870 (Breceptin)	Dual B1R and B2R antagonist	Phase I	Small cell lung cancer	No information available.	[17]

ACE, angiotensin-converting enzyme; B1R, bradykinin B1 receptor; B2R, bradykinin B2 receptor; NCT, National Clinical Trial (http://www.clinicaltrials.gov).

## 3. Kinin Receptors in Disease

Kinin receptor polymorphism, expression and activation have been shown to be associated with both protective (i.e., protection of the endothelium) and deleterious effects (i.e., inflammation, mediated infection and pain) depending on disease setting. This dichotomy has led to the development of B1R and B2R agonists and antagonists for indications such as cardiovascular, renal and airway diseases, in addition to acute pain, neurological disorders and cancers [1,41,42]. Table 1 lists B1R and B2R agonists and antagonists that have evaluated in clinical testing. In this section, we will provide an overview of diseases where B1R/B2R expression and activation have been implicated.

### 3.1. Pain

Kinins, and especially bradykinin, are involved in pain regulation and hyperalgesia after tissue injury and inflammation. Although they differ in expression level, B1R and B2R stimulation activates similar physiological responses. B2R plays a role in the acute inflammatory response, while B1R is induced by the accumulation of des-Arg^9^-BK and mediates long-term inflammation and inflammatory pain [6]. B1R and B2R antagonists are being studied as analgesic agents, such as MEN16132 for knee pain in osteoarthritis and MK-0686 for postoperative dental pain. B2R antagonists target the early inflammatory phase while B1R antagonists are more appropriated for the treatment of chronic inflammation. B2R may be involved in spinal and supraspinal nociceptive neurotransmission in the central nervous system (CNS) [6]. There is also growing evidence that suggests B1R and B2R antagonists may be efficient for treating other types of pain in the absence of inflammation including neuropathic pain responses [43,44].

### 3.2. Cardiovascular Diseases

Among the two receptors, B2R is more associated with cardiovascular disorders [1,45]. The activation of B2R by kinins can induce vasodilatation, plasma extravasation and cardioprotective effects (anti-hypertrophic, anti-proliferative and anti-atherosclerotic efficacy) [46,47]. B2R agonists have therefore been proposed for preventing and treating cardiovascular disorders like hypertension, ischemic heart disease, left ventricular hypertrophy, vascular remodeling and congestive heart failure [46]. On the other hand, B2R activation is linked with inflammation. In this context, B2R antagonists effectively treat the inflammation-associated cardiovascular diseases [1]. The activation of B1R can confer protection against cardiac ischemia [1]; however, proinflammatory cytokines can stimulate B1R for leukocyte recruitment and/or activation, which is involved in disorders such as atherosclerosis [48]. B1R expression is also associated with hypertension and heart failure [49]. Therefore, both B1R agonists and antagonists are being viewed as therapeutic options.

### 3.3. Renal Diseases

The severity of renal diseases has been shown to be negatively correlated to the level of excreted tissue kallikreins [1]. Activation of B1R and B2R seems to inhibit renal oxidative stress and inflammation and protect endothelial cells function and decrease fibrosis, but the exact mechanism remains to be further investigated [18,50]. These findings therefore suggest that activation of the KKS can improve renal outcomes in various renal diseases especially diabetic nephropathy [51,52]. In diabetic nephropathy, while inconsistent results have been reported for B1R, B2R polymorphism has been reported and is associated with the risk of developing the disease [52].

### 3.4. Neurological Disorders

Kinins are mediators in the CNS and are involved in neurological disorders; therefore, central kinin receptors are potent targets. As with other diseases described previously, B1R and B2R exhibit protective and/or adverse effects in these pathologies [53]. For example, B1R antagonists have been proposed as anti-epileptic agents, while B1R agonists and B2R antagonists could be used for stroke [54]. Kinin receptors are also involved in other neurological disorders including traumatic brain injury, spinal cord injury, Alzheimer’s disease and multiple sclerosis [53,55,56]. B1R and B2R agonists have also been suggested in combination treatment for brain tumors such as glioma to enhance delivery of anti-cancer drugs [57].

### 3.5. Cancers

Increased generation of kinins and B1R and B2R expression have been reported in cancer [17,58]. Of the two, B1R expression is considered the more ideal target because of its low expression in normal tissues. B1R activation has been shown to potentiate malignant behaviors by inducing cell proliferation, migration, and angiogenesis [59,60,61]. The pro-inflammatory effect of B1R activation can modulate the tumor microenvironment, priming for distant metastasis [62]. The overexpression of B1R has been observed in a large spectrum of malignancies including esophageal, cervical, gastric, prostate, lung, renal cancers [17]. As a consequence, B1R antagonists have been studies as therapeutic agents in preclinical settings with promising results [63,64].

### 3.6. Other Indications

Other diseases have been reported to be associated with kinin receptors. Aberrant B1R and B2R expression profile has been observed in inflammatory arthritis, preeclampsia, gastrointestinal diseases and hereditary angioedema [41,65,66,67,68].

## 4. Imaging of Kinin Receptors

We performed a search on PubMed using the following query: bradykinin receptor OR B1R OR B2R AND imaging. The abstracts of the potentially relevant articles were screened (208 entries; Figure 4). The references of the eligible articles were searched for relevant publications. Studies that measured B1R or B2R expression in vitro (e.g., immunohistochemistry), used protein-tagged B1R or B2R, or described indirect methods for measuring bradykinin response (e.g., calcium imaging) were excluded from analyses. A search of the Molecular Imaging and Contrast Agent Database (MICAD) yielded no entries for bradykinin receptor.

Of the imaging agents that are described in literature, those explored for in vivo applications generally target B1R instead of B2R. The primary reason is the expression of B2R in normal tissues, which intrinsically limits the achievable image contrast (uptake in target of interest vs. in background tissues). This limitation affects B1R less as it is pathologically induced. In terms of imaging probes, modified kinins at the *N*-terminal (i.e., fluorophore, radioisotope) have been the most studied. The use of an endogenous sequence as a template for developing imaging agents is common for GPCRs [70].

### 4.1. B1R Imaging Agents

The Marceau group is one of the early research groups to use fluorescently tagged peptides to study bradykinin receptors [71,72]. Bawolak et al. synthesized three 5(6)-carboxyfluorescein-tagged peptides—B-10372, B-10376, and B-10378—to characterize possible agonist-induced receptor translocation [72]. B-10372 and B-10376 are based on the potent antagonist B-9958 (Lys-Lys-Arg-Pro-Hyp-Gly-CpG-Ser-D-Tic-CpG) [73]. B-10372 has no linker, while B-10376 has an ε-aminocaproyl linker that separates the carboxyfluorescein from the peptide sequence. B-10378 is the agonist Lys-des-Arg^9^-BK with the identical fluorophore and linker at the *N*-terminal. Conjugation with the fluorophore reduced binding affinity to the B1R; however, the peptides were able to clearly image recombinant B1R expression in a transfected cell model (Figure 5). While the antagonist (B-10376) labeled the plasma membrane, the agonist (B-10378) showed discontinuous plasma membrane labeling, minimal internalization and partial caveolae/lipid raft mediated endocytosis.

As B1R is reported to be upregulated in pro-inflammatory and oxidative-stress settings, Talbot and co-workers conjugated a BODIPY fluorophore to des-Arg^9^-BK (BdABK) to study the role of B1R in diabetic polyneuropathy [74]. Rats were treated with streptozotocin (STZ) to induce diabetes and euthanized four days following STZ injection. The thoracic spinal cord was harvested for confocal microscopy. Colocalization studies using BdABK suggested that B1R is expressed in the microglial cells, astrocytes and sensory C fibers within the thoracic spinal cord. The in vivo activity of BdABK was further demonstrated by its ability to induce thermal hyperalgesia in STZ-treated mice. This study further elucidates the mechanism of neuropathic pain in diabetes, and implicates B1R as a potential target for pharmacological intervention.

Yeo et al. conjugated fluorescein isothiocyanate (FITC) to Lys-des-Arg^9^-BK and Lys-[D-Phe^8^]des-Arg^9^-BK [75]. These two peptides were used to detect inflammation in living cells on a slide chip. Briefly, A549 cells were cultured onto a slide chip that consisted of a Starna flow cell and polydimethylsiloxane walls. This set-up enables the continuous flow of culture medium under CO_2_ and temperature-controlled environment. After starvation, cells were treated with *Pseudomonas aeruginosa* lysate to induce inflammation. Cells were stained with the FITC-conjugated peptides and analyzed by optical microscopy (Figure 6). Higher fluorescence signal was detected in inflamed cells than in normal cells. According to the authors, the peptides showed better differentiation between the two groups of cells than using an antibody approach targeting the Toll-like-receptor 1. 

Charest-Morin and Marceau generated a series of fusion proteins based on the agonist des-Arg^9^-BK and antagonist [Leu^8^]des-Arg^9^-BK, respectively [76]. For their design, the peptide sequence was cloned behind a fluorescent protein (enhanced GFP (EGFP) or mCherry) and a variable spacer peptide (Arg-Gln)_n_. HEK293a cells were transfected with the expression vectors and lysed to collect the fusion proteins. Because of how they are produced, the targeting sequences are restricted to natural amino acids. The fusion proteins were evaluated for their ability to displace [^3^H]Lys-des-Arg^9^-BK in competition binding assays, and to label B1R in fluorescence microscopy. The best candidate, EGFP-(Asn-Gly)_15_-Lys-des-Arg^9^-BK, was able to compete with [^3^H]Lys-des-Arg^9^-BK, but with 10-fold less binding affinity than unmodified Lys-des-Arg^9^-BK. The uptake of EGFP-(Asn-Gly)_15_-Lys-des-Arg^9^-BK was predominantly membranous, confirming that B1R is not subjected to agonist-induced receptor translocation unlike B2R [1]. The fusion protein did not bind to HEK 293a expressing B2R or angiotensin converting enzyme, demonstrating specificity.

In 2012, our research group began developing B1R radiotracers for cancer imaging with PET [77,78,79,80,81,82,83,84]. We synthesized three kallidin derivatives and appended the 1,4,7,10-tetraazacyclododecane-1,4,7,10-tetraacetic acid (DOTA) chelator at *N*-terminus, separated by a 6-aminohexanoic acid (Ahx) or 9-amino-4,7-dioxanonanoic acid (dPEG2) linker [77]. The radiotracers were radiolabeled with ^68^Ga-gallium (t_1/2_: 68 min), an isotope suited to the fast pharmacokinetics of short peptides [85,86]. ^68^Ga-P03083 (^68^Ga-DOTA-Ahx-[Leu^9^,des-Arg^10^]kallidin) retains the native sequence, while ^68^Ga-SH01078 (^68^Ga-DOTA-Ahx-[Hyp^4^,Cha^6^,Leu^9^, desArg^10^]kallidin) and ^68^Ga-P03034 (^68^Ga-DOTA-dPEG2-[Hyp^4^,Cha^6^,Leu^9^,desArg^10^]kallidin) contain unnatural amino acids to improve stability. ^68^Ga-P03083 was ineffective at visualizing B1R-positive tumors, unless co-administered with phosphoramidon, a peptidase inhibitor (Figure 7). ^68^Ga-SH01078 and ^68^Ga-P03034 showed higher tumor uptake, minimal accumulation in normal tissues, and rapid renal clearance. ^68^Ga-P03034 achieved peak tumor uptake of 2.19 ± 1.08%ID/g at 1 h post-injection (p.i.). This study demonstrated that non-invasive B1R imaging was feasible and underscored the significance of metabolic stability.

Our research group investigated the effect of linkers on the biodistribution and tumor uptake of B1R imaging agents [78]. Based on the sequence ^68^Ga-DOTA-linker-Lys-Arg-Pro-Hyp-Gly-Cha-Ser-Pro-Leu, ^68^Ga-SH01078, ^68^Ga-P03034, ^68^Ga-P04115, and ^68^Ga-P04168 were synthesized with Ahx, dPEG2, Gly-Gly and 4-amino-(1-carboxymethyl)piperidine (Pip) linker, respectively. Based on competition binding assays, the K_i_ values of the peptides ranged from 3.6–27.8 nM. In mice bearing hB1R-expressing HEK293T tumors, tracers accumulated in tumor with minimal background activity except for kidneys and bladder. ^68^Ga-P04168, which had the best affinity, generated the highest tumor uptake (4.15 ± 1.13%ID/g at 1 h p.i.) and lowest background activity, leading to a >2-fold improvement in contrast (tumor-to-blood, tumor-to-muscle, and tumor-to-liver ratios) over the other three candidates. It was concluded the linker selection can influence binding affinity, pharmacokinetics, and tumor targeting.

At this point in time, all of the reported B1R radiotracers leveraged an antagonist sequence. We were interested in studying whether the use of an agonist could lead to improved pharmacokinetics and tumor uptake. Consequently, we synthesized ^68^Ga-Z01115 (^68^Ga-DOTA-Ahx-Lys-Arg-Pro-Hyp-Gly-Cha-Ser-Pro-D-Phe), which is nearly identical to ^68^Ga-SH01078 except that for a Leu^9^/D-Phe^9^ substitution [79]. This substitution was sufficient for changing the agonist/antagonist properties of the peptide [87], which was validated by calcium release assays. The binding affinity of ^68^Ga-Z01115 for B1R was 25.4 nM, which was similar to previously reported ^68^Ga-SH01078. For in vivo studies, we observed an increase in B1R+ tumor uptake (5.65 ± 0.59%ID/g at 1 h p.i.), and tumor-to-background ratios. Although PET tracers are administered at microdose levels and unlikely to exert pharmacological effects [88], we thought it prudent to forego the use of agonists and avoid the risk of hyperalgesia.

We focused our attention to identifying other peptide antagonists in the literature with enhanced metabolic stability that can be repurposed for imaging [73,89]. We proceeded to radiolabel two antagonists, B9858 (Lys-Lys-Arg-Pro-Hyp-Gly-Igl-Ser-D-Igl-Oic) and B9958 [80,81,82]. Both sequences contain four unnatural amino acids to confer resistance to peptidases. In addition to ^68^Ga-gallium, we radiolabeled these peptides with ^18^F-fluorine (t_1/2_: 110 min) using Al^18^F chelation [90] or an ^18^F-^19^F isotope exchange reaction on the trifluoroborate motif [91,92,93]. ^18^F-fluorine has a slightly longer physical half-life, better decay properties and emission energy, which leads to better resolution [94,95]. The peptides exhibited nanomolar binding affinity for B1R (K_i_ values: 0.1–5.5 nM). The highest uptake in hB1R-expressing HEK293T tumors was observed for ^68^Ga-Z02176 (28.9 ± 6.21%ID/g at 1 h p.i.), as shown in PET images (Figure 8). The tumor-to-blood and tumor-to-muscle ratios for ^68^Ga-Z02176 were 56.1 ± 17.3 and 167 ± 57.6, respectively. These results represented a major advance in the development of B1R PET imaging agents.

Given the success observed with B9858 and B9958, we decided to investigate the sequence of R954 (Ac-Orn-Arg-Oic-Pro-Gly-αMePhe-Ser-D-2-Nal-Ile) [83]. R954 is reportedly more stable than B9858 against peptidase degradation [96]. Furthermore, a phase I clinical study of R954 was approved based on favorable pharmacokinetics, stability, and safety profile [97]. These observations served as the impetus for selecting R954 as a lead for developing B1R imaging agents. We synthesized ^68^Ga-labeled (^68^Ga-HTK01083) and ^18^F-labeled (^18^F-HTK01146) derivatives of R954, both employing a dPEG2 linker. While high contrast PET images were obtained (Figure 9), the uptake of ^68^Ga-HTK01083 and ^18^F-HTK01146 in B1R+ tumors (8.46 ± 1.44 and 9.25 ± 0.69%ID/g) was lower than the values obtained with B9858 or B9958. We hypothesized that this is due to reduced binding affinity of the R954 derivatives towards B1R (24.8–30.5 nM), which was about 10–12-fold higher than that of ^68^Ga-Z02137 (2.5 nM). Table 2 summarizes the development of B1R radiopharmaceuticals based on kinins.

As mentioned, research groups have developed small molecule inhibitors to target kinin receptors for improved oral bioavailability and brain penetrance. We synthesized two B1R radiotracers, ^18^F-Z02035 and ^18^F-Z02165, based on a series of 2-[2-[[(4-methoxy-2,6-dimethylphenyl)sulfonyl]methylamino]ethoxy]acetamide derivatives reported by Barth et al. [98]. This was accomplished by replacing the *N*-alkyl group with a radioactive 2-fluoroethyl group [84]. ^18^F-Z02035 and ^18^F-Z02165 had nanomolar affinity (0.9–2.8 nM) for B1R. However, when evaluated in vivo, the uptake in B1R+ HEK293T tumors were 3.77 ± 0.79 and 3.21 ± 0.79%ID/g at 1 h p.i., respectively. High uptake was observed in liver and gastrointestinal tract for both tracers, which reflected the lipophilic nature of the compounds. ^18^F-Z02035 exhibited some defluorination as bone uptake was noted, while ^18^F-Z02165 showed off-target binding to thyroid. Collectively, these results suggested that peptides are more appropriate targeting vectors.

**Table 2 pharmaceuticals-13-00199-t002:** Comparison of binding affinity, hydrophilicity, tracer uptake, and uptake ratios of selected B1R-targeting peptides. Adapted from [82,83].

Peptide Name	Peptide Sequence	K_i_ (nM)	Average Tissue Uptake (1 h p.i., %ID/g)		Average B1R+ Tumor-to-Background Contrast Ratio (1 h p.i.)		Ref
			B1R+ Tumor	Kidney	To Blood	To Muscle	
**Bradykinin**	Arg-Pro-Pro-Gly-Phe-Ser-Pro-Phe-Arg	5.7 ^a^					[99]
**Kallidin**	Lys-Arg-Pro-Pro-Gly-Phe-Ser-Pro-Phe-Arg	7.4 ^a^					[99]
**[Leu^9^,desArg^10^]kallidin**	Lys-Arg-Pro-Pro-Gly-Phe-Ser-Pro-Leu	8.9 ^b^					[99]
^68^Ga-P03083	^68^Ga-DOTA-Ahx-[Leu^9^,desArg^10^]kallidin	2.6 ± 0.7	0.79 ± 0.22	4.95 ± 0.86	10.4 ± 3.78	8.18 ± 1.69	[77]
^68^Ga-SH01078	^68^Ga-DOTA-Ahx-[Hyp^4^,Cha^6^,Leu^9^,desArg^10^]kallidin	27.8 ± 4.9	2.06 ± 0.52	3.14 ± 0.62	7.78 ± 2.20	30.2 ± 7.42	[77]
^68^Ga-P03034	^68^Ga-DOTA-dPEG2-[Hyp^4^,Cha^6^,Leu^9^,desArg^10^]kallidin	16 ± 1.9	2.17 ± 0.49	4.50 ± 2.17	5.72 ± 2.20	25.5 ± 13.1	[77]
^68^Ga-P04115	^68^Ga-DOTA-Gly-Gly-[Hyp^4^,Cha^6^,Leu^9^,desArg^10^]kallidin	11 ± 2.5	1.96 ± 0.83	4.02 ± 2.40	6.37 ± 3.82	26.1 ± 8.91	[78]
^68^Ga-P04168	^68^Ga-DOTA-Pip-[Hyp^4^,Cha^6^,Leu^9^,desArg^10^]kallidin	3.6 ± 0.2	4.15 ± 1.13	4.02 ± 1.22	15.9 ± 6.84	78.1 ± 28.5	[78]
^ 68 ^ Ga-Z01115	^68^Ga-DOTA-Ahx-[Hyp^4^,Cha^6^,D-Phe^9^,desArg^10^]kallidin	25.4 ± 5.1	5.65 ± 0.59	4.63 ± 1.27	24.4 ± 12.9	82.9 ± 35.0	[79]
**B9858**	Lys-Lys-Arg-Pro-Hyp-Gly-Igl-Ser-D-Igl-Oic	10.1 ^b^					[89]
^68^Ga-P04158	^68^Ga-DOTA-dPEG2-B9858	1.5 ± 1.9	19.6 ± 4.50	69.2 ± 7.39	19.2 ± 8.21	66.1 ± 17.0	[80]
^18^F-L08064	^18^F-AmBF_3_-Mta-Pip-B9858	0.1 ± 0.1	3.94 ± 1.24	36.2 ± 5.78	6.69 ± 3.60	21.3 ± 4.33	[81]
**B9958**	Lys-Lys-Arg-Pro-Hyp-Gly-Cpg-Ser-D-Tic-Cpg	0.089					[100]
^68^Ga-Z02090	^68^Ga-DOTA-dPEG2-B9958	1.1 ± 0.8	14.1 ± 1.63	50.1 ± 9.68	29.9 ± 5.58	124 ± 28.1	[80]
^68^Ga-Z02176	^68^Ga-DOTA-Pip-B9958	2.5 ± 0.8	28.9 ± 6.21	90.9 ± 22.8	56.1 ± 17.3	167 ± 57.6	[82]
^68^Ga-Z02137	^68^Ga-NODA-Mpaa-Pip-B9958	2.6 ± 0.7	14.0 ± 4.86	85.2 ± 12.1	34.3 ± 15.2	103 ± 30.2	[82]
^18^F-L08060	^18^F-AmBF_3_-Mta-Pip-B9958	0.5 ± 0.3	4.20 ± 0.98	30.9 ± 6.74	14.7 ± 3.56	48.6 ± 10.7	[81]
^18^F-Z04139	Al^18^F-NODA-Mpaa-Pip-B9958	1.4 ± 0.7	22.6 ± 3.41	101 ± 14.4	58.0 ± 20.9	173 ± 42.9	[82]
**R954**	Ac-Orn-Arg-Oic-Pro-Gly-αMePhe-Ser-D-2Nal-Ile	10.0 ± 3.1					[83]
^68^Ga-HTK01083	^68^Ga-DOTA-dPEG2-R954	30.5 ± 7.6	8.46 ± 1.44	66.1 ± 9.70	6.32 ± 1.44	20.7 ± 3.58	[83]
^18^F-HTK01146	^18^F-AmBF_3_-Mta-dPEG2-R954	24.8 ± 2.8	9.25 ± 0.69	77.0 ± 19.5	7.24 ± 2.56	19.5 ± 4.29	[83]

^a^ pIC_50_, ^b^ pEC_50_. DOTA: 1,4,7,10-tetraazacyclododecane-1,4,7,10-tetraacetic acid; Ahx: aminohexanoic acid; Hyp: hydroxyproline; Cha: cyclohexylalanine; dPEG2: 9-amino-4,7-dioxanonanoic acid; Igl: alpha-(2-indanyl)glycine; Cpg: cyclopentylglycine; AmBF_3_-Mta: 4-(N-trifluoroborylmethyl-N,N-dimethylammonio)methyl-1,2,3-triazole-1-acetic acid; D-Tic: D-tetrahydroisoquinoline-3-carboxylic acid; Pip: 4-amino-(1-carboxymethyl)piperidine; Mpaa: 4-methylphenylacetic acid; Orn: ornithine; Oic: octahydroindole-2-carboxylic acid; αMePhe: N-methylphenylalanine; NODA: 1,4,7-triazacyclononane-1,4-diacetate; D-2-Nal: 3-(β-naphthyl)-D-alanine.

### 4.2. B2R Imaging Agents

Given the important role of B2R in both physiological and pathological phenomenon, efforts to visualize receptor activity through imaging agents were explored. The first example of B2R-targeted fluorescence imaging was performed with B-10380, a B2R antagonist (B-9430: D-Arg-[Hyp^3^,Igl^5^,D-Igl^7^,Oic^8^]bradykinin) with a carboxyfluorescein and an ε-aminocaproic acid linker at its *N*-terminus [71]. While B-10380 suffered a loss of affinity as compared to the parent B-9430, treatment of HEK293a cells transiently transfected with B2R with B-10380 led to labelling of the cell membrane. The specificity of B-10380 was determined by the use of non-transfected cells and a blocking control via LF 16-0687, another B2R antagonist. The authors attempted to use a biotinylated B-9430 derivative (B-10330) to enable streptavidin-Alexafluor 594 binding for live-cell microscopy; however, no fluorescence signal was observed.

Gera et al. designed two fluorescent agonists by conjugating a carboxyfluorescein with and without an ε-aminocaproic acid linker to the native bradykinin sequence [101]. While the carboxyfluorescein-based bradykinin showed an approximate 3000-fold loss in binding as compared to bradykinin, the carboxyfluorescein-ε-aminocaproyl-bradykinin had a 400-fold loss in affinity. The latter fluorescent agent was still able to label HEK293a cells transfected with B2R, but was unable to bind to a truncated B2R variant that lacked the G-protein receptor kinase domain. Tracking of the fluorescent signal showed an endosomal-mediated internalization route, which was expected from an agonist sequence. The previously reported B-10380 was also included as an antagonist counterpart in this study, and was able to label both the whole and truncated B2R receptor in transfected cells.

With the understanding that the *N*-terminus of bradykinin-based agonists and antagonists are tolerant of extensions, Gera et al. created several bifunctional conjugates by linking various functional or fluorescent moieties at the peptide’s *N*-terminus, with or without a ε-aminocaproic acid linker [102]. This study aimed at both assessing the fluorescence imaging of B1R, B2R and ACE and the capability of agonists to internalize various cargoes. B2R agonists included the original bradykinin sequence, maximakinin and B-9972 (D-Arg-Arg-Pro-Hyp-Gly-Igl). B-9430 was used as the sole B2R antagonist. Functional moieties included chlorambucil, biotin, pentafluorocinnamic acid and ferrocene. Evaluation with in vitro radioligand competition binding assays showed that the conjugation of fluorescent moieties such as carboxyfluorescein and Alexafluor 350 lowered their affinity for B2R, though the presence of the linker mitigated this decrease. Cell-based microscopy experiments showed variable behavior between the agonist- and antagonist-based fluorescent agents, consistent with their purported identity. While the antagonist-based fluorescent agents aggregated at the extracellular membrane, the agonist-based agents were internalized via beta-arrestin recruitment into endosomes, which was supported by colocalization fluorescent experiments using labeled beta-arrestin and endosomal markers.

Charest-Morin et al. successfully leveraged agonist-induced internalization of B2R to deliver protein cargoes intracellularly [103]. The protein sequence of maximakinin, a bradykinin homologue, was subcloned at the *C*-terminus of EGFP and expressed in HEK293a cells. EGFP was chosen because its high molecular weight and steric bulk helped simulate a large cargo. Maximakinin was selected over bradykinin as the targeting sequence, as it was predicted it would be less affected affinity-wise because of its length (19-mer vs. 9-mer). Fluorescence imaging with the labeled maximakinin showed signal co-localization with beta-arrestin labeled with mCherry in cells transfected with B2R. The specificity was further confirmed with quantitative cytofluorometry analysis and with anatibant, a B2R antagonist, as a blocking control. The labeled maximakinin was further shown to be more specific than carboxyfluorescein-labeled bradykinin, with no labeling of cells expressing ACE, most likely due to steric interactions.

To take advantage of the tissue penetrance and reduced autofluorescence of NIR imaging, Gera and co-workers conjugated a cyanine-based NIR dye (Cy7) at the *N*-terminus of B-9430, an antagonist, and B-9972 (D-Arg-[Hyp^3^,Igl^5^,Oic^7^,Igl^8^]bradykinin), an agonist, yielding B-10665 and B-10666 [104]. The binding affinity of B-10665 for B2R (K_i_: 25.7 nM) was 1.5-fold lower compared to B-9430, while B-10666 (K_i_: 598 nM) suffered a loss of approximately 35-fold in affinity to B2R compared to B-9972. Microscopy of B2R+ and B1R− HEK293a cells showed that B-10665 labeled the membrane surface of B2R+ HEK293 cells at concentrations as low as 10 nM (Figure 10). The uptake can be blocked with the use of icatibant. As for B-10666, NIR signal was internalized in B2R+ HEK293a cells. This signal was also blockable in the presence of icatibant. Neither B-10665 nor B-10666 showed any signal in B2R− HEK293a cells. Although not investigated in this study, the high sensitivity of B-10665 opens up the possibility for in vivo animal studies.

The only instance of a radiolabeled imaging agent targeting B2R utilized icatibant/HOE140 (D-Arg-Arg-Pro-Hyp-Gly-Thi-Ser-D-Tic-Oic-Arg) as a targeting vector with a linker and chelator placed at its *N*-terminus [105]. With a benzyl-based linker and a cyclam-based 4-[(1,4,8,11-tetraazacyclotetradec-1-yl)-methyl]benzoic acid (CTPA) chelator, Stahl and co-workers showed that this conjugated peptide (CTPA-HOE140) retained high affinity to B2R (8.03 nM vs. 1.07 nM for the unmodified peptide). CTPA-HOE140 was complexed with ^99m^Tc, a SPECT radioisotope, and assessed in healthy rats to study its biodistribution in vivo. ^99m^Tc-CTPA-HOE140 showed rapid clearance from circulation over a period of 4 h (13.2%ID/g at 5 min p.i., 5.5%ID/g at 30 min p.i., and 0.5%ID/g at 4 h p.i.), and approximately equal uptake in the liver and kidneys (5.0%ID/g and 4.3%ID/g at 4 h p.i., respectively), two major organs governing the metabolism and excretion. While it had relatively low uptake in other healthy tissues, no SPECT imaging and follow-up investigations were performed, clouding the potential of this radiotracer.

## 5. Perspective and Summary

The signaling of kinins and kinin-based receptors plays a key physiological role in maintaining homeostasis and a pathological role in mediating inflammatory processes in certain diseases. Both B1R and B2R are prominent therapeutic targets. MI techniques can elucidate the molecular events leading to phenotypic disease and enable the visualization of target engagement of novel therapeutics. Many imaging agents targeting B1R or B2R have been developed to date, using kinin-based agonists and antagonists. With respect to B2R, development of imaging agents has been confined to primarily cell-based microscopy, though the development of NIR and SPECT imaging agents opens up possibilities for in vivo investigations. MI of B1R has shown greater success than its counterpart, with the development of several high-contrast PET radiotracers that can use either ^68^Ga or ^18^F, two commonly available PET radioisotopes. These PET agents can be used for patient stratification to identify those with diseases (e.g., inflammation, cancer etc.) that express high levels of B1R and are suitable for B1R-targeted therapies. The same imaging agent can also be used to monitor and evaluate treatment response. It should be noted that more studies are needed before these probes can enter the clinic, including comprehensive toxicity and dosimetry studies [88]. Finally, given their interactions with the immune system, the development and translation of B1R and B2R imaging agents will benefit from animal models that better recapitulate human pathophysiology (e.g., humanized systems, nonhuman primates).

## Figures and Tables

**Figure 1 pharmaceuticals-13-00199-f001:**
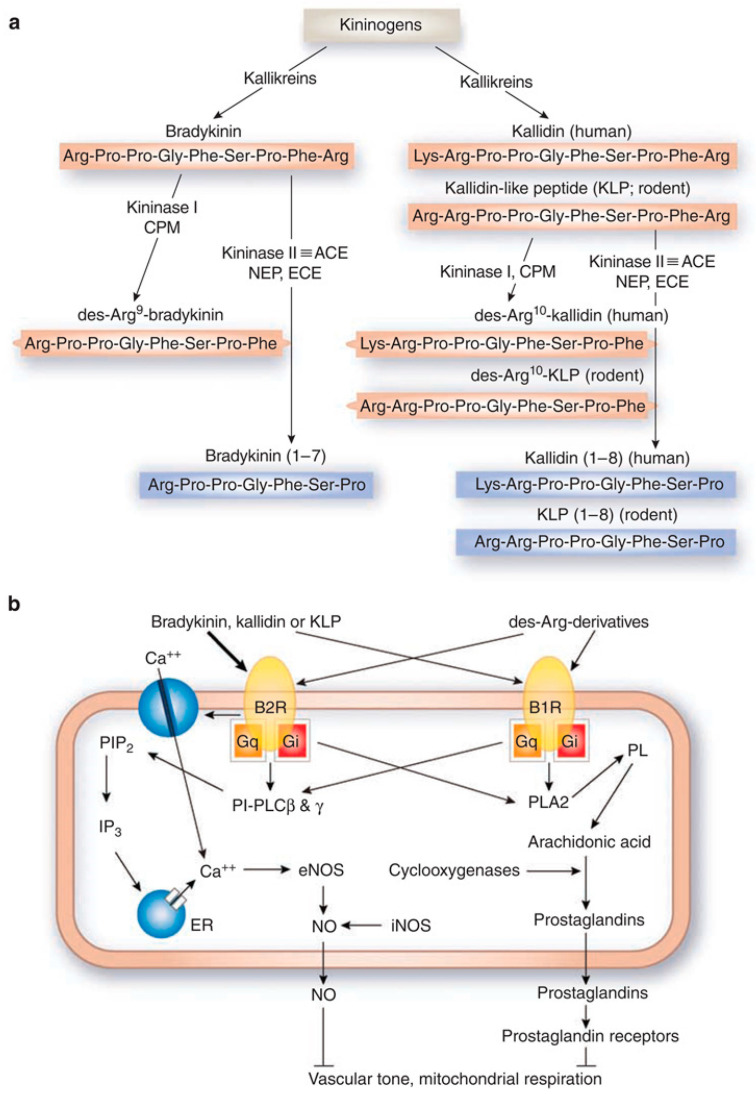
Components and signaling of the kallikrein–kinin system. (**a**) Biosynthesis and metabolism of kinins. Kininogens are processed by kallikreins to produce bradykinin and kallidin. Kininase I removes the carboxyl-Arg from bradykinin and kallidin to yield des-Arg^9^-bradykinin and des-Arg^10^-kallidin, respectively. Enzymatic cleavage by kininase II (angiotensin I-converting enzyme), neprilysin, and endothelin-converting enzyme can inactivate kinins. CPM, carboxypeptidase-M; ACE, angiotensin I-converting enzyme; NEP, neprilysin (endopeptidase 24.11); ECE, endothelin-converting enzyme; red, active peptides; blue, inactive peptides. (**b**) Binding of kinins to bradykinin receptors and intracellular signaling. The thickness of arrows arising from the kinins indicates the relative potency of each peptide to elevate intracellular calcium concentrations. PIP2, phosphatidylinositol-4,5-bisphosphate; PI-PLC, phosphatidylinositol-specific phospholipase C; IP3, 1,4,5-inositol triphosphate; ER, endoplasmic reticulum; PL, phospholipids; PLA2, phospholipase A2; NO, nitric oxide; eNOS, endothelial NO synthase; iNOS, inducible NO synthase. Figure reproduced from Kakoki, M., et al. [18].

**Figure 2 pharmaceuticals-13-00199-f002:**
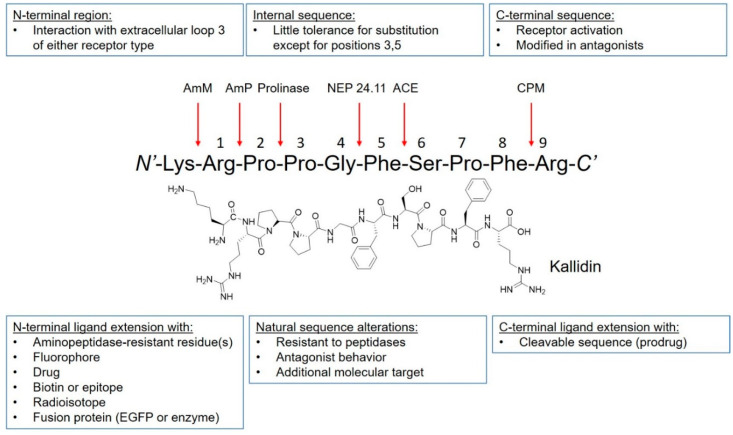
Peptide plasticity of kinins. Kinin peptides can be modified to improve bradykinin B1R or bradykinin B2R selectivity, increase resistance to enzymatic cleavage, confer agonistic or antagonistic properties, and prolong pharmacologic effect. Moreover, peptide ligands can be conjugated to fluorophores, drugs, radioisotope complex, and be fused with other proteins. Arrows indicate positions of potential cleavage. AmM: aminopeptidase M; AmP: aminopeptidase P; NEP 24.11: neutral endopeptidase 24.11; ACE: angiotensin-converting enzyme; CPM: carboxypeptidase M. Figure adapted from Marceau, F., et al. [25].

**Figure 3 pharmaceuticals-13-00199-f003:**
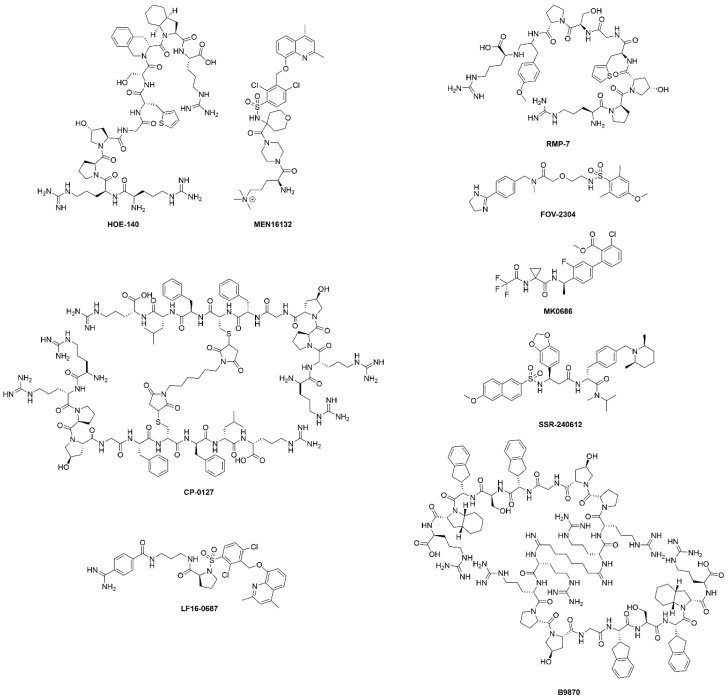
Chemical structures of B1R/B2R agonists and antagonists listed in Table 1. Both peptides and small molecules have been evaluated in clinical trials. The chemical structure of BI-113823, B1R antagonist, has not been disclosed.

**Figure 4 pharmaceuticals-13-00199-f004:**
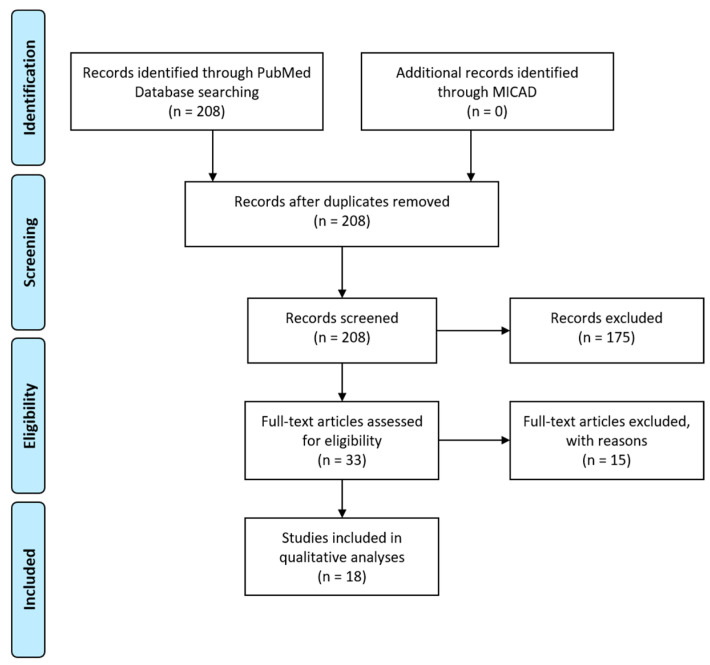
Literature search for B1R and B2R imaging agents. Flow diagram to illustrate the process of article selection of this systematic review. Template redrawn from Moher, D.; et al. [69].

**Figure 5 pharmaceuticals-13-00199-f005:**
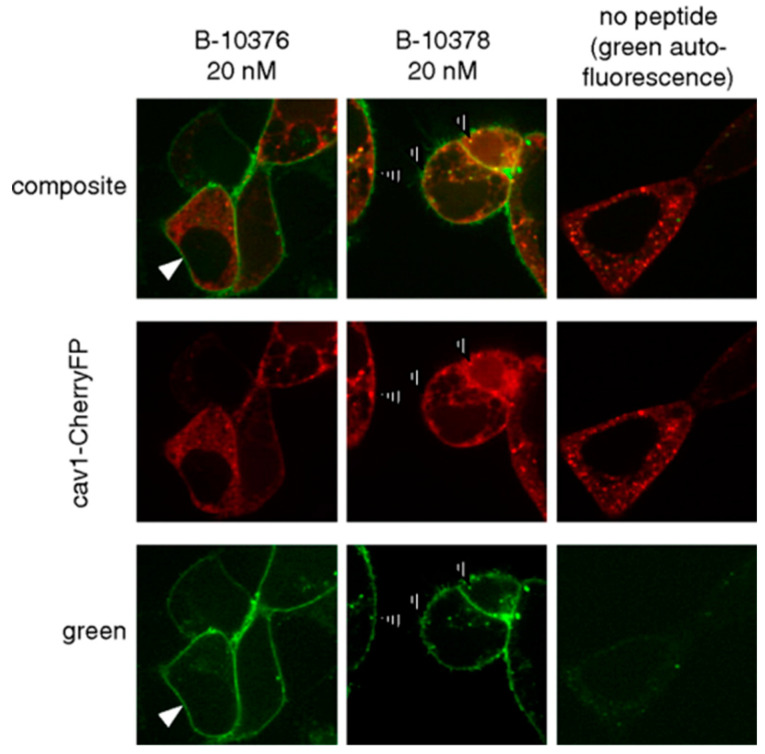
Confocal fluorescence imaging in HEK293a cells transiently expressing rabbit B1R and cavelolin-1-CherryFP. Cells were treated with 20 nM B-10376 (antagonist) or B-10378 (agonist). Both peptides labeled plasma membrane, but cells treated with B-10378 displayed enhance punctate fluorescence condensation that colocalized with cavelolin-1 indicated by hashed arrows. Figure reproduced from Bawolak, M.T.; et al. [72].

**Figure 6 pharmaceuticals-13-00199-f006:**
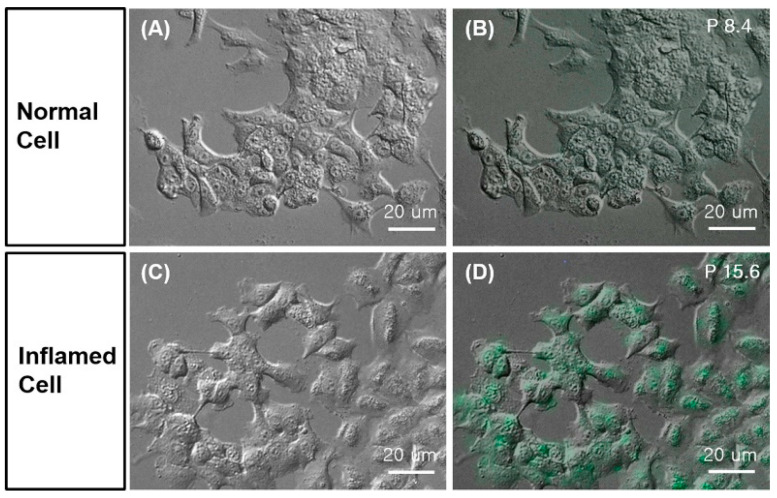
B1R inflammation imaging. After 1-day starvation, A549 cells were treated with 100 µg *P. aeruginosa* lysate for 4 h to induce inflammation. Cells were stained with 100 nM of FITC-conjugated Lys-[D-Phe^8^]des-Arg^9^-BK for 1 h at 37°C and analyzed with optical imaging. Differential interference contrast (DIC) microscope image of (**A**) normal and (**C**) inflamed cells. Fluorescence and DIC merged images of (**B**) normal and (**D**) inflamed cells. Figure reproduced from Yeo, K.B.; et al. [75].

**Figure 7 pharmaceuticals-13-00199-f007:**
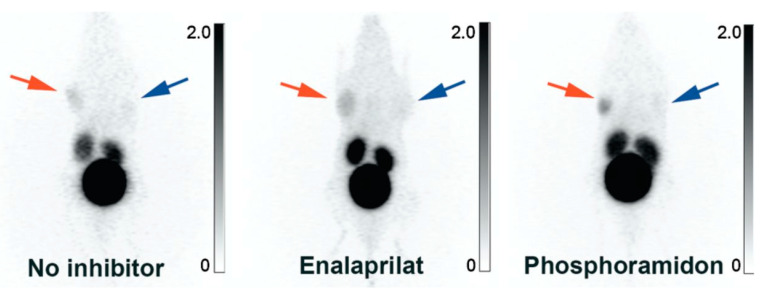
B1R positron emission tomography imaging. Maximum intensity projection PET images obtained with ^68^Ga-P03083 without (**left**), with enalaprilat (**middle**), and with co-injection of phosphoramidon (**right**). The B1R+ tumor (red arrow) was located on the right shoulder (the animal is viewed on a coronal projection, ventral viewpoint). The B1R− tumor (blue arrow) had no significant uptake. The gray scale bar to the right of each image is set in units of %ID/g. Figure reproduced from Lin, K.S.; et al. [77].

**Figure 8 pharmaceuticals-13-00199-f008:**
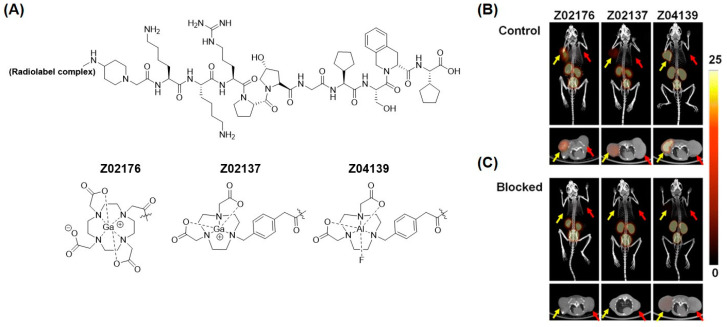
Optimization of B1R radiopharmaceuticals. (**A**) Chemical structure of B1R radioligands. (**B**) Representative 1 h post-injection MIP (maximum intensity projection) and axial PET/CT images of radiolabeled Z02176, Z02137, and Z04139 in mice bearing both B1R+ (indicated by yellow arrows) and B1R– (indicated by red arrows) tumors without and (**C**) with co-injection of 100 μg cold standard. The displacement in tumor uptake demonstrates target specificity of the imaging agent. The scale bar is in units of %ID/g. Figure reproduced from Zhang, Z.; et al. [82].

**Figure 9 pharmaceuticals-13-00199-f009:**
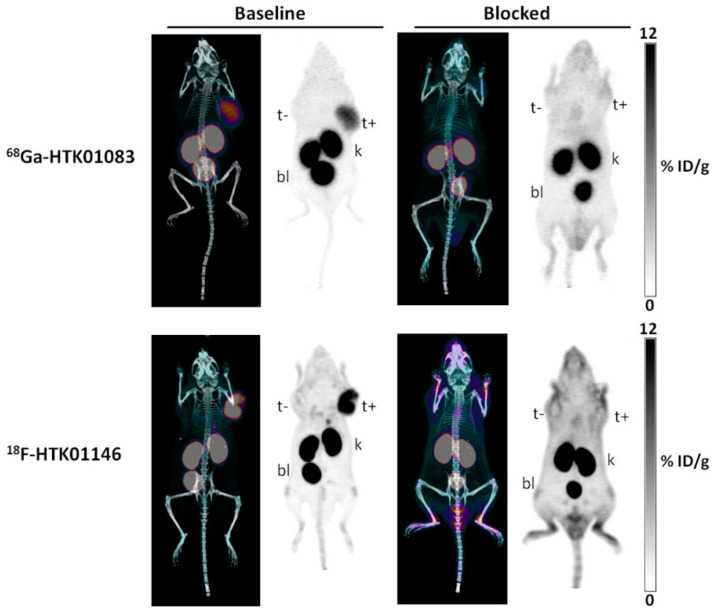
Maximal intensity projection images of PET/CT and PET with ^68^Ga-HTK01083 and ^18^F-HTK01146 in mice bearing HEK293T and HEK293T::hB1R tumors at 1 h post-injection. Blocking studies were performed with co-injection of 100 μg of R954. t+, B1R+ tumor; t−, B1R- tumor; k, kidney; bl, bladder. Figure adapted from Kuo, H.T.; et al. [83].

**Figure 10 pharmaceuticals-13-00199-f010:**
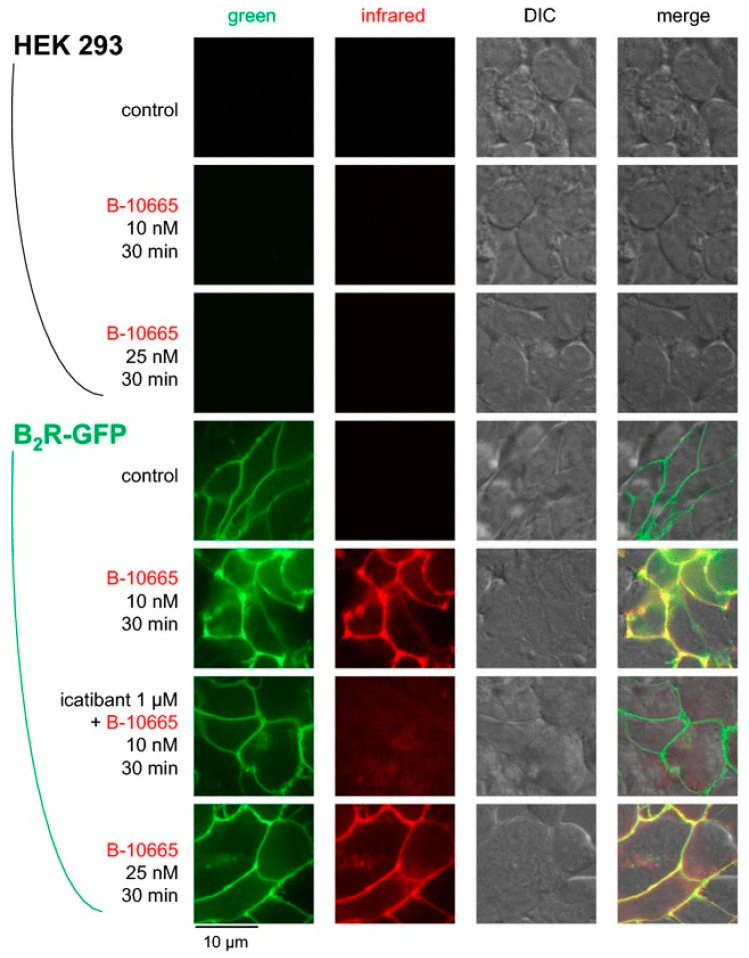
Near-infrared imaging of B2R. Imaging of HEK 293a cells expressing B2R-GFP using Cy7-conjugated B-10665 (infrared fluorescence rendered as red false color). Labelling was performed for 30 min at 37 °C, followed by rinse. Original magnification 95×. Figure reproduced with permission from Infrared-emitting, peptidase-resistant fluorescent ligands of the bradykinin B2 receptor: application to cytofluorometry and imaging

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
