# Peer review of "A Systematic Review of Molecular Imaging Agents Targeting Bradykinin B1 and B2 Receptors"

_pharmaceuticals, 2020, doi:10.3390/ph13080199_

Round 1

Reviewer 1 Report

This manuscript reviewed molecular imaging agents targeting bradykinin receptors. The manuscript was well written and would be of special interest to the readers of Pharmaceuticals.

Comments:

  1. Adding structure of small molecule agonists and antagonists in Table 1 will be better.
  2. Table 1 is hard to see because there is no space between each study. Please add space or line.
  3. In page 11, the authors described that a Phase I clinical study of R954 was approved. Please add target indication of the clinical study.

Reviewer 2 Report

This is a well-written and interesting review article on the possible role of molecular imaging agents targeting bradykinin receptors.

The manuscript is well-organized and tables are informative.

I would suggest to add a figure reporting the literature search process and article selection as usually recommended in systematic reviews. 

Reviewer 3 Report

This is a comprehensive review on the development of molecular imaging (MI) agents targeting Kinin B1 and B2 receptors, using Kinin-based agonists and antagonists. Authors made valuable contributions in this field over the last five years. Although this part of the review is well addressed, other parts need further updating and additional minor points deserve careful attention:

1-  P.2, para 2 and Figure 1: Kininase II is also named angiotensin-1 converting enzyme. Kininase II is not a group that includes ACE, neprilysin, and endothelin-converting enzyme. While kininase II/ACE is a dipeptidyl carboxypeptidase, the other enzymes are endopeptidases. Kininase II = ACE is the conventional nomenclature.

2- Table 1: This table includes very helpful and relevant information. However, one can wonder if anything new has been reported on the therapeutic use of B1R/B2R agonists and antagonists since 2014 (reference 10 by da Costa et al.). An update is required in the frame of this review.

3- P. para 3: Kinin receptors in disease: In this section, authors mentioned for each disease the reference (1) by Leeb-Lundberg et al. which was published 15 years ago. Many seminal contributions and reviews on these disorders/pathologies have been written since then, particularly on neuropathic pain. This section has to be updated.

4- Figure 3 legend: read 20 nM and not 20 mM. Read cavelolin (line 4).

5- P. 10, para 3: ‘We focused our attention to identifying other peptide agonists in literature’. Authors want to mean ‘peptide antagonists’ and not ‘agonists’.

6- Figure 6: Add a better identification of Graph C in figure legend.

7- P. 11, line 6 from bottom: Reference 65 is not from Barth et al. Be careful! Reference 65 by Regoli et al. in Table 2 is correct.

 8- The cellular localization of B1R was shown in thoracic spinal cord of type 1 diabetic rats by confocal microscopy with the use of a fluorescent agonist, [Nalpha-Bodipy]-des-Arg9-BK (BdABK) (J Neuroinflammation. 2009, 6:11. doi: 10.1186/1742-2094-6-11). This is probably the sole preclinical-study using a fluorescent marker labeling a B1R agonist.

P. 7, para 4.1: read Bawolak and not Bawolek

Author Response

Reviewer 3:

This is a comprehensive review on the development of molecular imaging (MI) agents targeting Kinin B1 and B2 receptors, using Kinin-based agonists and antagonists. Authors made valuable contributions in this field over the last five years. Although this part of the review is well addressed, other parts need further updating and additional minor points deserve careful attention:

  1. 2, para 2 and Figure 1: Kininase II is also named angiotensin-1 converting enzyme. Kininase II is not a group that includes ACE, neprilysin, and endothelin-converting enzyme. While kininase II/ACE is a dipeptidyl carboxypeptidase, the other enzymes are endopeptidases. Kininase II = ACE is the conventional nomenclature.

Response: Thank you for pointing out the mistake. We have revised the statement to read, “Kinins are also enzymatically regulated by kininase II (angiotensin I-converting enzyme), neprilysin, and endothelin-converting enzyme.”

  1. Table 1: This table includes very helpful and relevant information. However, one can wonder if anything new has been reported on the therapeutic use of B1R/B2R agonists and antagonists since 2014 (reference 10 by da Costa et al.). An update is required in the frame of this review.

Response: Using the clinicaltrials.gov database, we did not see any new compounds that have entered the clinic beyond those listed by da Costa et al. We updated the entries (clinical phases, status and obtained results). Notably:

  1. HOE-140: Completed Phase III and now has entered Phase IV for ACE-inhibitor associated angioedema and heart failure. The Phase II study for mitochondria and chronic kidney disease did not yield any therapeutic efficacy.
  2. MEN16132: The Phase II study initiated in 2014 is now complete and results are provided.
  3. RMP-7: The Phase I study for HIV infection and cryptococcal meningitis was not mentioned in da Costa’s review and was added.
  4. SSR-240612: We realized that this compound was listed in the 2014 review, but was missing from the Table 1. It has been added in the revised version.

  1. para 3: Kinin receptors in disease: In this section, authors mentioned for each disease the reference (1) by Leeb-Lundberg et al. which was published 15 years ago. Many seminal contributions and reviews on these disorders/pathologies have been written since then, particularly on neuropathic pain. This section has to be updated.

Response: We reduced updated citations as suggested. Notably, we have added the following citations in Section 3:

  1. Kinin receptors in disease

Tang, M.; He, F.; Ma, L.; Liu, P.; Wang, J.; Zhu, X. Bradykinin Receptors in Ischemic Injury. Curr. Neurovasc. Res. 2018, 15, 359–366, doi:10.2174/1567202616666181123151629.

  1. Pain

Choi, S.I.; Hwang, S.W. Depolarizing effectors of bradykinin signaling in nociceptor excitation in pain perception. Biomol. Ther. 2018, 26, 255–267, doi:10.4062/biomolther.2017.127.

Cernit, V.; Sénécal, J.; Othman, R.; Couture, R. Reciprocal regulatory interaction between TRPV1 and kinin B1 receptor in a rat neuropathic pain model. Int. J. Mol. Sci. 2020, 21, doi:10.3390/ijms21030821.

  1. Cardiovascular diseases

Manolis, A.J.; Marketou, M.E.; Gavras, I.; Gavras, H. Cardioprotective properties of bradykinin: role of the B2 receptor. Hypertens. Res. 2010, 33, 772–777, doi:10.1038/hr.2010.82.

Sriramula, S. Kinin B1 receptor: A target for neuroinflammation in hypertension. Pharmacol. Res. 2020, 155, 104715, doi:10.1016/j.phrs.2020.104715.

  1. Renal diseases

Alhenc-Gelas, F.; Bouby, N.; Girolami, J.-P. Kallikrein/K1, Kinins, and ACE/Kininase II in Homeostasis and in Disease Insight From Human and Experimental Genetic Studies, Therapeutic Implication. Front. Med. 2019, 6, doi:10.3389/fmed.2019.00136.

  1. Neurological disorders

Mandadi, S.; Leduc-Pessah, H.; Hong, P.; Ejdrygiewicz, J.; Sharples, S.A.; Trang, T.; Whelan, P.J. Modulatory and plastic effects of kinins on spinal cord networks. J. Physiol. 2016, 594, 1017–1036, doi:10.1113/JP271152.

Caetano, A.L.; Dong-Creste, K.E.; Amaral, F.A.; Monteiro-Silva, K.C.; Pesquero, J.B.; Araujo, M.S.; Montor, W.R.; Viel, T.A.; Buck, H.S. Kinin B2 receptor can play a neuroprotective role in Alzheimer’s disease. Neuropeptides 2015, 53, 51–62, doi:10.1016/j.npep.2015.09.001.

  1. Cancers

Liu, Y.S.; Hsu, J.W.; Lin, H.Y.; Lai, S.W.; Huang, B.R.; Tsai, C.F.; Lu, D.Y. Bradykinin B1 receptor contributes to interleukin-8 production and glioblastoma migration through interaction of STAT3 and SP-1. Neuropharmacology 2019, 144, 143–154, doi:10.1016/j.neuropharm.2018.10.033.

Dubuc, C.; Savard, M.; Bovenzi, V.; Lessard, A.; Côté, J.; Neugebauer, W.; Geha, S.; Chemtob, S.; Gobeil, F. Antitumor activity of cell‐penetrant kinin B1 receptor antagonists in human triple‐negative breast cancer cells. J. Cell. Physiol. 2019, 234, 2851–2865, doi:10.1002/jcp.27103.

Zhou, Y.; Wang, W.; Wei, R.; Jiang, G.; Li, F.; Chen, X.; Wang, X.; Long, S.; Ma, D.; Xi, L. Serum bradykinin levels as a diagnostic marker in cervical cancer with a potential mechanism to promote VEGF expression via BDKRB2. Int. J. Oncol. 2019, 55, 131–141, doi:10.3892/ijo.2019.4792.

  1. Other diseases

Kleine, S.A.; Budsberg, S.C. Synovial membrane receptors as therapeutic targets: A review of receptor localization, structure, and function. J. Orthop. Res. 2017, 35, 1589–1605, doi:10.1002/jor.23568.

Dutra, R.C. Kinin receptors: Key regulators of autoimmunity. Autoimmun. Rev. 2017, 16, 192–207, doi:10.1016/j.autrev.2016.12.011.

  1. Figure 3 legend: read 20 nM and not 20 mM. Read cavelolin (line 4).

Response: We corrected the typos.

  1. 10, para 3: ‘We focused our attention to identifying other peptide agonists in literature’. Authors want to mean ‘peptide antagonists’ and not ‘agonists’.

Response: The reviewer is correct. We meant to write, “peptide antagonists” and not “agonists”. Thank you for pointing out this error.

  1. Figure 6: Add a better identification of Graph C in figure legend.

Response: Figure 6 is now Figure 8. The caption now reads, “Optimization of B1R radiopharmaceuticals. (A) Chemical structure of B1R radioligands. (B) Representative 1 h post-injection MIP (maximum intensity projection) and axial PET/CT images of radiolabeled Z02176, Z02137, and Z04139 in mice bearing both B1R+ (indicated by yellow arrows) and B1R– (indicated by red arrows) tumors without and (C) with co-injection of 100 μg cold standard. The displacement in tumor uptake demonstrates target specificity of the imaging agent. The scale bar is in units of %ID/g. Figure adapted with permission from Mol Pharm. 2016, 13(8):2823-2832.

  1. 11, line 6 from bottom: Reference 65 is not from Barth et al. Be careful! Reference 65 by Regoli et al. in Table 2 is correct.

Response: Thank you for pointing out the error. The citation has been corrected (Barth et al. From bradykinin B2 receptor antagonists to orally active and selective bradykinin B1 receptor antagonists. J. Med. Chem. 2012, 55, 2574–2584, doi:10.1021/jm2016057).

  1. The cellular localization of B1R was shown in thoracic spinal cord of type 1 diabetic rats by confocal microscopy with the use of a fluorescent agonist, [Nalpha-Bodipy]-des-Arg9-BK (BdABK) (J Neuroinflammation. 2009, 6:11. doi: 10.1186/1742-2094-6-11). This is probably the sole preclinical-study using a fluorescent marker labeling a B1R agonist.

Response: We have added the following paragraph to include the study, “As B1R is reported to be upregulated in pro-inflammatory and oxidative-stress settings, Talbot and co-workers conjugated a BODIPY fluorophore to des-Arg9-BK (BdABK) to study the role of B1R in diabetic polyneuropathy [74]. Rats were treated with streptozotocin (STZ) to induce diabetes, and euthanized four days following STZ injection. The thoracic spinal cord was harvested for confocal microscopy. Colocalization studies using BdABK suggested that B1R is expressed in the microglial cells, astrocytes and sensory C fibers within the thoracic spinal cord. The in vivo activity of BdABK was further demonstrated by its ability to induce thermal hyperalgesia in STZ-treated mice. This study further elucidates the mechanism of neuropathic pain in diabetes, and implicates B1R as a potential target for pharmacological intervention.” ​

  1. 7, para 4.1: read Bawolak and not Bawolek.

Response: We corrected the spelling.

Reviewer 4 Report

The paper presents the first systematic review of scientific literature on imaging agents targeting bradykinin B1 and B2 receptors. Design and development of imaging agents for bradykinin receptors is an underexplored area of research, despite their potential use in drug development for these biological targets. The manuscript is well written, sufficiently detailed, and can serve as an introductory reference on the subject. As a shortcoming, the discussion about therapeutic perspectives of B1R/B2R as a drug target and about specific role of the successful imaging agents could have been more concrete, providing a clearer perspective/guidance for their use.

Author Response

The paper presents the first systematic review of scientific literature on imaging agents targeting bradykinin B1 and B2 receptors. Design and development of imaging agents for bradykinin receptors is an underexplored area of research, despite their potential use in drug development for these biological targets. The manuscript is well written, sufficiently detailed, and can serve as an introductory reference on the subject. As a shortcoming, the discussion about therapeutic perspectives of B1R/B2R as a drug target and about specific role of the successful imaging agents could have been more concrete, providing a clearer perspective/guidance for their use.

Response: Thank you for the comment. We have added the following sentences into the Perspective and Summary section, “These PET agents can be used for patient stratification to identify those with diseases (e.g. inflammation, cancer etc.) that express high levels of B1R and are suitable for B1R-targeted therapies. The same imaging agent can also be used to monitor and evaluate treatment response. It should be noted that more studies are needed before these probes can enter the clinic including comprehensive toxicity and dosimetry studies [88].”